# Impact of Non-Invasive Physical Plasma on Heat Shock Protein Functionality in Eukaryotic Cells

**DOI:** 10.3390/biomedicines11051471

**Published:** 2023-05-18

**Authors:** Yanqing Wang, Alexander Abazid, Steffen Badendieck, Alexander Mustea, Matthias B. Stope

**Affiliations:** 1Department of Gynecology and Gynecological Oncology, University Hospital Bonn, Venusberg-Campus 1, 53127 Bonn, Germany; 2Department of General, Visceral and Thorax Surgery, Bundeswehr Hospital Berlin, Scharnhorststrasse 13, 10115 Berlin, Germany

**Keywords:** cold plasma, cold atmospheric plasma, cold atmospheric pressure plasma, tissue tolerable plasma, cellular stress

## Abstract

Recently, biomedical research has increasingly investigated physical plasma as an innovative therapeutic approach with a number of therapeutic biomedical effects. It is known from radiation and chemotherapy that these applications can lead to the induction and activation of primarily cytoprotective heat shock proteins (HSP). HSP protect cells and tissues from physical, (bio)chemical, and physiological stress and, ultimately, along with other mechanisms, govern resistance and treatment failure. These mechanisms are well known and comparatively well studied in drug therapy. For therapies in the field of physical plasma medicine, however, extremely little data are available to date. In this review article, we provide an overview of the current studies on the interaction of physical plasma with the cellular HSP system.

## 1. Introduction

Physical plasma describes a gas with a sufficient degree of ionization, which makes it conductive and capable of exhibiting a collective response to electromagnetic fields [1]. In addition to solid, liquid, and gas, physical plasma forms the fourth state of matter in physics. Physical plasma can be generated by microwaves, ionizing radiation, or high electric voltages [2]. For use in medicine, electrons are accelerated in an electric field so that they can impact atoms. This impact ionization leads to an avalanche charge multiplication, in which the free electrons are constantly accelerated in the electric field, and thus continue the process [3]. In addition to the generation of free electrons, ions, excited atoms and molecules, radicals and electromagnetic radiation, UVC radiation, for example, are also generated. In the atmosphere, some of the energy is also transferred to the particles in the ambient air so that energized oxygen particles are also formed, primarily reactive oxygen species (ROS) [4]. For application to patients, non-thermal physical plasmas are used with temperatures barely above human body temperature [1]. For this purpose, the physical plasmas are ionized only to a small extent, and free electrons are additionally stopped by dielectric materials. In this dielectric barrier discharge technology, the large, slow atoms and molecules are only slightly accelerated and the overall temperature of the physics plasma remains low [3]. Physical plasma administered in medical therapy is particularly gentle on tissues compared to other physical therapy methods such as ionizing radiation and laser. The biomedical effects are not induced by the introduction of high energies into the tissue, but are mainly transferred by biochemical mechanisms. The physically developed terms for such medically used physical plasmas such as ‘cold atmospheric plasma’ or ‘cold atmospheric pressure gas plasma’ do not indicate any medical properties. Therefore, the term ‘non-invasive physical plasma’ seems to be more suitable. This reflects the tissue-preserving properties of physical plasma and the initial confusion with blood plasma can be ruled out from the outset. In the following, the term ‘non-invasive physical plasma’ (NIPP) will therefore be used when referring to non-thermal physical plasma for medical use.

As early as 1996, it was demonstrated that NIPP can inactivate microorganisms very efficiently [4]. Subsequent studies have shown that a number of medically relevant pathogens, including multi-resistant bacteria, can be eradicated by treatment with NIPP. These include, for example, bacteria of the genera *Escherichia*, *Streptococcus*, *Staphylococcus* and *Pseudomona*, but also fungi and multicellular parasites [4,5,6]. Furthermore, NIPP is markedly beneficial to wound healing. The combination of antimicrobial and regenerative effects made NIPP an excellent therapeutic modality for the treatment of acute and chronic wounds including ulcers [7,8,9,10,11,12,13,14,15]. NIPP interacts with the ambient atmosphere resulting in the formation of ROS in the gas phase [16,17]. These diffuse into the aqueous extracellular space of tissues [18] and lead to a local increase in ROS concentration. The resulting redox stress leads to the oxidation of cellular lipids, proteins, and nucleic acids. This impairs cellular structures and limits their functions including, for example, ATP biosynthesis at the mitochondrial membrane [19]. Cells react to this with stress-induced responses such as the reduction of cell growth, cell motility, and induction of apoptosis [18,19,20]. Taken together, the cellular and molecular NIPP effects often lead to devitalizing effects that can be used to treat neoplasms, especially cancer [21,22]. Eukaryotic cells respond precisely to environmental influences, physiological stress, and therapeutic interventions. Chemical factors such as drugs, but also physical noxae—including, for instance, ionizing radiation—facilitate the expression of stress-induced cytoprotective proteins, primarily belonging to the heat shock protein (HSP) family. HSP are classified into six families based on their molecular weight [23]. HSP represent stress-induced cell survival factors involved in various cell responses (Figure 1).

In tumors, HSP are usually upregulated compared to non-malignant tissue areas and ensure the survival of the metabolically highly active cancer cells. Anticancer therapies typically further enhance HSP induction [24,25]. Thus, cancer cells protect themselves from the cytotoxic effects of the therapeutic agent and HSP-mediated resistance to the therapeutic agent occurs [26,27,28]. Accordingly, HSP inhibitors can be administered as an anticancer drug or to support chemotherapy [29]. Pharmacological inhibition of HSP70 in prostate cancer cells also leads to the downregulation of other HSP, revealing the regulatory interplay of different HSP family members in pathological processes [30]. Furthermore, HSP can suppress mechanisms of cell death such as necrosis and apoptosis [31,32,33], and buffer redox stress by ROS [34]. HSP functions have been implicated in numerous pathological processes beyond cancer, including ischaemic injury, inflammatory and infectious conditions, transplant complications, and immune disorders [35,36,37,38,39,40].

HSPs are also known to be involved in the immune response, which is believed to play a critical role in immune regulation by interacting with the immune system in several ways. For example, they are chaperones for antigenic peptides, which are fragments of proteins that are recognized by the immune system [41,42]. HSP can mediate the transport of these peptides to antigen-presenting cells, which are specialized immune cells that present the peptides to other immune cells, such as T-cells [43]. This process can stimulate an immune response to the antigenic peptide. In addition, HSP have been shown to have immunomodulatory effects on immune cells. For example, HSP can induce the maturation of dendritic cells, which are important antigen-presenting cells [44,45], and the activation of T-cells, which are crucial for the adaptive immune response [46]. Moreover, HSPs have been shown to have anti-inflammatory effects, which can help to prevent or reduce inflammation caused by the immune system [47]. Meanwhile, studies have shown that NIPP can modulate the activity of immune cells in several ways. For example, NIPP has been shown to increase the migration of immune cells, such as macrophages and neutrophils, to the site of inflammation or injury [48,49,50]. Additionally, NIPP also stimulates the production of cytokines and chemokines, which are signaling molecules that play important roles in the immune regulatory network [51,52]. These molecules can help to attract immune cells to the site of injury or infection and activate them to eliminate pathogens or damaged tissue [51,52]. NIPP has also been shown to induce apoptosis in certain types of immune cells, such as T- and B-cells [53,54].

In accordance with these important physiological and pathological HSP functions, it seems important to also consider potential influences on HSP expression and functionality after NIPP exposition. NIPP has been proven by many studies promoting tissue regeneration, wound healing, and cell repair [55,56]. ROS has been proved by many studies to have an adverse effect on cell survival and cause a stress response in cells [57,58], which can be produced in large quantities by NIPP [4,16,17]. However, NIPP was demonstrated to promote wound healing and cell proliferation and protect against cell damage. In tumor cells, NIPP can selectively kill tumor cells, and HSP have been proven to be highly expressed in most tumor cells and are thus related to their prognosis and malignancy. As a cytoprotective protein, HSP are also a marker protein of malignant tumors. Therefore, it is of great significance to characterize the effect of NIPP on HSP expression in cells after NIPP exposure using various NIPP devices (Table 1).

## 2. HSP27

HSP27 is encoded by the HSPB1 gene located on human chromosome 7. The HSP27 gene contains three exons encoding 205 amino acids [68]. The structure of HSP27 mainly includes an α-crystallin domain, a WDPF domain, a partially conserved domain amino-terminal sequence, and a flexible carboxyl-terminal domain [69,70,71]. In contrast to large HSP, HSP27 is ATP-independent [69]. The phosphorylated forms of HSP27 are present in the nucleus and cytoplasm with a concomitant degradation of cytoplasmic and nuclear proteins [72], which may involve different cellular signaling networks.

The phosphorylation of HSP27 promotes the formation of small oligomers, whereas dephosphorylation promotes the formation of large oligomers, and this process is dynamically reversible [73]. It is widely believed that strong phosphorylation of three serine sites (Ser_15_, Ser_78_, and Ser_82_) present in the N-terminal domain induces dissociation of large HSP27 oligomers [74,75]. Under physiological conditions, when HSP27 is not phosphorylated, HSP27 exists as large oligomers (up to 1000 kDa) with molecular chaperone functions including refolding of unfolded proteins, regulation of cytoskeletal dynamics, and involvement in cell cycle regulation [76,77]. Under cellular stimulation conditions (high temperature, chemical toxins, radiation, etc.), HSP27 can be phosphorylated and activated by the p38 mitogen-activated protein kinase (p38 MAPK) pathway [26,78,79,80], protein kinase B (PKB) [81], protein kinase C (PKC) [82], protein kinase D (PKD) [83,84], protein kinase G (PKG), PKC-beta/ERK1/2, and PKC-beta/p38 MAPK [80,81], causing conformational changes and dissociation of oligomers into smaller HSP27 aggregates. Furthermore, it is controlled by various factors, such as tumor necrosis factor-α (TNFα) [85], transforming growth factor-β (TGFβ) [86,87], insulin-like growth factor-1 (IGF1) [88], and steroid hormones [89,90]. Studies have shown that HSP27 expression, which is not upregulated under oxidative stress, reduces intracellular ROS levels [91]. It also supports reduced forms of glutathione and mitochondrial membrane potential. Furthermore, it consolidates intracellular redox homeostasis by reducing intracellular iron levels and thus generating hydrogen radicals through the Fenton reaction, the acid oxidation with hydrogen peroxide catalyzed by iron salts. HSP27 stimulates glucose 6-phosphate dehydrogenase (G6PD) activity through the interaction of this reductase with its small and highly phosphorylated oligomers. Thus, the presence of HSP27 strongly attenuated protein oxidation, DNA damage, lipid peroxidation, and cytoskeletal structure disruption. The signaling mechanism of HSP27 phosphorylation does not depend on ROS but depends on the redox state. Pro-inflammatory cytokines induce the formation of ROS and reduce HSP27 content by promoting the formation of peroxynitrite, ultimately leading to retinal capillary endothelial cell apoptosis [92]. Schmidt et al. showed that in epithelial HaCaT keratinocytes, MAP kinases and ROS-associated HSP27, a downstream effector of the p38 signaling cascade, showed a clear correlation of phosphorylation level with NIPP treatment time. Similar to p38, the maximal phosphorylation level was reached after 180 s of treatment. Compared with H_2_O_2_-treated cells, phosphorylation of HSP27 was only slightly stimulated [93,94]. Debora et al. demonstrated that HSP27 release was enhanced upon the NIPP treatment of LNCaP and PC-3 cells [66,94]. After NIPP treatment, increased release of HSP27 was also found in OVCAR3 cells [67]. Furthermore, HSP27 is also involved in cytoskeleton remodeling and cell migration, which is modulated by NIPP treatment [59]. Furthermore, the two immune cell lines CD4+ T helper cell line Jurkat and monocyte cell line THP-1 were exposed to NIPP, and THP-1 cells were less sensitive to NIPP treatment in a pro-apoptotic and pro-proliferative manner than Jurkat cells [65]. Activation of HSP27 was detectable in THP-1 monocytes. The authors speculated that one of the reasons why THP-1 cells are more resistant to NIPP treatment than Jurkat cells might be the expression of cytoprotective HSP27. HSP27 activation, however, may play a key role in cellular escape from apoptosis after NIPP treatment.

## 3. HSP40

HSP40 topology contains a highly conserved J-domain responsible for binding to HSP70 and stimulating ATPase activity as a co-chaperone. HSP40 proteins are divided into three groups: Class A (DNAJA), Class B (DNAJB), and Class C (DNAJC). Class A and B consist of an N-terminal J domain, a Gly-Phe-rich region, two C-terminal β-barrel domains for substrate-binding, and a dimerization domain. The J-domain of HSP40 can bind to the HSP70 ATPase domain, thereby stimulating the ATPase activity of HSP70 [95].

HSP40 is involved in protein translation, folding, unfolding, refolding, stabilization, and thus degradation [95,96]. HSP40 levels are elevated in human head and neck cancers, which correlates with decreased overall survival [97]. Parrales et al. [98] found that HSP40 also interacts with and stabilizes misfolded p53. Knockdown of HSP40 triggers ubiquitin ligase-mediated proteasomal degradation of mutated p53, thereby reducing malignant features of cancer cells and representing tumor-promoting efficacy. Co-expression of HSP70 and HSP40 prevents loss of mitochondrial membrane potential and follows apoptosis in RAW 264.7 cells [99]. In human lymphoma cells, U937, expression levels of HSP40 and Bcl-2 associated athanogene (BAG3) were significantly higher in cells exposed to MIPP compared to controls [61]. HSP40 and BAG3 are co-chaperones of HSP70 and are mainly regulated by heat shock transcription factor 1 (HSF1), an anti-apoptotic factor induced by various types of stress [100,101].

## 4. HSP60

HSP60 is located on chromosome 2 and is encoded by the gene HSPD1 [102]. HSP60 protein is mainly located in mitochondria and is required to maintain the integrity and function of the mitochondrial respiratory chain and cell survival [103,104]. HSP60 interacts with the accessory chaperone HSP10 to correct the folding of nascent proteins, restore the structure of misfolded proteins, and maintain the steady state of mitochondrial proteins [105]. Mitochondrial functionality appears to be the main activity of HSP60. Moreover, the protein has few functions in other typically HSP-dependent cell responses and HSP60 expression appears to be regulated very precisely [30,106,107]. Low levels of HSP60 are scattered in the cytoplasm, cell membrane surface, cell-derived exosomes, extracellular space, and the bloodstream [108,109]. Here, HSP60 is thought to be involved in membrane transport and signal transduction. In the absence of ATP, HSP60 exists in the form of a stable heptameric monocyclic ring. When ATP binds, HSP60 proteins form bicyclic structures followed by binding to HSP10 [105]. ATP hydrolysis can cause conformational changes in the apical domain and drive the folding and release of the bound protein [105,110,111]. Upregulation of HSP60 is considered an indicator of mitochondrial stress, such as increased mitochondrial ROS and mitochondrial DNA damage [112]. HSF1, a key regulator of heat shock response, controls HSP60 expression [113,114]. Due to dual subcellular localization, the function of HSP60 as an anti- and pro-apoptotic factor may depend on its cellular location and its ability to shuttle between mitochondria and cytosol [115,116,117]. Overexpression of mitochondrial HSP60 was able to inhibit doxorubicin-induced cardiomyocyte apoptosis by increasing anti-apoptotic Bcl-xL and Bcl-2, decreasing pro-apoptotic Bax, and inhibiting procaspase-3 activation. Among them, HSP60 not only interacts with Bax and Bcl-xL but also inhibits ubiquitination of Bcl-xL [118]. Loss of HSP60 is associated with altered levels of many mitochondrial proteins, including increased expression of pro-apoptotic Bax and decreased expression of anti-apoptotic Bcl-2 [119,120,121]. Furthermore, in cancer cells, mitochondrial HSP60 acts as a regulator of mitochondrial permeability transition by binding to cyclophilin D (CypD). HSP60 depletion induces CypD-dependent mitochondrial permeability transition, leading to apoptosis [122]. Following NIPP treatment, these liquids are known as NIPP-treated media and have been shown to have cytotoxic effects on cancer cells. Tornin et al. demonstrated that NIPP increases HSP60 expression by increasing H_2_O_2_ in cancer cells, and HSP60 expression was inhibited with catalase inhibitors and pyruvate [60]. They found that NIPP-treated media had greater cytotoxicity with loss of antitumor selectivity.

## 5. HSP70

The sequence and protein structure of HSP70 are highly conserved in all species examined. All HSP70 isoforms contain two functional domains: the N-terminal nucleotide-binding domain (NBD) and the C-terminal substrate-binding domain (SBD). The factor regulates various mechanisms of cellular protein processing, including folding of nascent and misfolded proteins, protein assembly, transport, degradation, and prevention and disassembly of protein aggregates. This chaperone activity of HSP70 is achieved through an ATP-regulated cycle of polypeptide substrate binding and release. HSP70 have two intrinsic activities: first, as an ATPase, and second, to bind polypeptide substrates. These two activities are tightly coupled, and the chaperone activity strictly depends on this allosteric coupling. The chaperone activity of HSP70 is aided by two well-characterized accessory chaperones: Hsp40s and nucleotide exchange factors (NEF) [123]. HSP40 accelerates ATP hydrolysis by HSP70, while NEF mediate the regeneration of ADP by ATP. Taken together, HSP70, HSP40, and NEF constitute the most important chaperone mechanisms for protein folding and protein stabilization in cells [124,125].

HSP70 blocks apoptotic pathways by interacting with significant signaling proteins upstream, downstream, and within mitochondrial regulatory processes. Upstream of mitochondria, HSP70 binds and blocks c-Jun N-terminal kinase (JNK) activity. HSP70 inhibits apoptosis in a caspase-dependent mechanism by inhibiting JNK. For example, in hypertonicity-induced apoptosis, HSP70 deficiency induces JNK activation and caspase-3 activation [126]. HSP70 has also been shown to bind and stabilize non-phosphorylated PKC and Akt [127]. At the mitochondrial level, HSP70 inhibits Bax translocation and insertion into the mitochondrial outer membrane. Thus, HSP70 prevents mitochondrial membrane permeabilization and the release of cytochrome c and apoptosis-inducing factor (AIF) [128]. At the mitochondrial level, HSP70 blocks coupled Bax translocation with HSP40, thereby preventing mitochondrial outer membrane permeability and inhibiting the release of cytochrome c and other mitochondrial apoptotic molecules such as AIF. HSP70 also acts at the post-mitochondrial level. It has been consistently found that HSP70 inhibits the release of cytochrome c downstream of apoptosis and upstream of caspase-3 activation. Indeed, HSP70 has been shown to bind directly to Apaf-1, thereby preventing the recruitment of procaspase-9 to apoptotic bodies [129,130].

In human lymphoma cells U937, NIPP treatment resulted in a significant increase in apoptosis. Expression levels of HSP40 and BAG3 were significantly enhanced [96,131], suggesting that NIPP exposure expands ROS levels inhibiting NIPP-induced apoptosis. Previous studies have shown that nitric oxide (NO) is produced during NIPP treatment and rapidly converted to other ROS, including NOX [132,133]. NO induces HSF1-regulated HSP70 expression and subsequent cytoprotection [125,126,134].

## 6. HSP90

Similar to HSP70, HSP90 is also a highly conserved ATP-dependent chaperone that binds to client proteins to enable their appropriate folding [135]. Structurally, this HSP90 is a dimer. The monomeric subunits contain the four domains N-terminal dimerization domain (NTD), intermediate domain (MD), charged variable length region (CR), and C-terminal domain (CTD) [23,136]. HSP90 exists in several isomers sharing high degrees of homology. These isoforms exist both intracellularly and extracellularly. Five functional HSP90 members have been identified in mammalian cells, including the major cytoplasmic isoforms HSP90α1/α2 and HSP90β, as well as glucose-regulated protein 94 (GRP94), and tumor necrosis factor receptor-associated protein 1 (TRAP1) in the endoplasmic reticulum and mitochondria, respectively [135]. The two cytoplasmic isoforms of HSP90, HSP90α, and HSP90β, share 85% sequence identity. HSP90β is constitutively expressed under normal physiological conditions. However, HSP90α is stress-regulated, and elevated levels of HSP90α are associated with poor cancer prognosis [136,137]. HSP90 is also liberated into the extracellular environment through an exosomal pathway distinct from the classical secretion pathway.

Under non-stress conditions, the master regulator HSF-1 is inactive and binds to HSP90-containing protein complexes. Hypoxia, hyperthermia, trauma, and metabolic stress conditions disrupt the HSP90-HSF-1 complex, prompting the formation of HSF-1 homotrimers that translocate to the nucleus and stimulate rapid transcription of heat shock genes, and HSP90 expression increases [138]. HSF-1 is modified by post-translational regulation, and its transcriptional activation depends on the state of hyperphosphorylation. When sufficient levels of cytosolic HSP90 are reached and the denatured protein is metabolized or reconfigured, the HSP90 binding site again becomes available to stabilize HSF-1 in an inactive form [139,140].

HSP90 has more than 300 client proteins, including protein kinases, transcription factors, oncoproteins and tumor suppressors. Nima et al. showed the aggregation of platelets treated with NIPP and methylcellulose loaded with polyethyleneimine-polypyrrole nanoparticles. The treatment of burn wounds with a compound system can significantly increase the expression level of HSP90 and promote wound healing [62,64]. Schmidt et al. showed that HSP90A, HSP90AB, and HSP90B were significantly up-regulated after treatment of epithelial keratinocyte HaCaT cells with NIPP [59]. Another study showed that NIPP-treated cancer cells (colon, prostate, breast) not only led to the cleavage of HSP90 but also related to the degradation of PKD2. Their results demonstrate that HSP90 cleavage following NIPP treatment causes degradation of its client protein PKD2 and leads to impaired proliferation and increased cancer cell death in cancer cells (Figure 2). Interestingly, the level of cell death following NIPP treatment was comparable to that achieved by lentivirus-mediated knockdown of HSP90 [63].

HSP are regulated by heat shock transcription factor 1 (HSF1) and serve as stress-induced cytoprotective factors. As little as 20 min incubation of cells at 39 °C activates the heat shock element-binding activity of HSF1 [141,142]. HSF1, however, is also activated by oxidative stress [143], as also occurs during exposure to NIPP. Thus, the presence of NO and NOx in cells leads to HSF1-dependent induction of HSP70 [144,145,146]. This is accompanied by protein stabilization and refolding as well as modulation of immune processes. The physiological effects depend on the HSP superfamily members involved and are likely to be cell type and tissue specific. In addition, the NIPP device applied must be considered. The NIPP technology used and the specific technical characteristics of the device determine the properties of the NIPP, thus affecting the concentrations and composition of the ROS formed and, ultimately, their biological effects. The existing studies on the interaction of NIPP and HSP systems indicate that HSP plays a significant role. However, current studies still need to answer numerous questions to paint a generalized picture.

## 7. Conclusions

Current evidence on the functionality of HSP clearly demonstrates that they have important functions in cell protection and survival. This is evident in the inhibition of apoptosis, but also in controlling other cellular responses such as proliferation, motility, protein stability, degradation and turnover. NIPP-induced modulation of these HSP functions, therefore, has a major impact on the viability of cells and tissues.

As with other therapeutic modalities, HSP are potential targets in NIPP therapy (Figure 2). The mostly observed induction of HSP expression could be a cell response to NIPP-induced (redox) stress and may contribute to cell survival. Degradation of HSP90, however, also shows that NIPP action can inactivate HSP and suppress entire downstream signaling and effector cascades. Data published so far on the anticancer effects of NIPP demonstrate that cancer cells can be eradicated depending on the duration of treatment. Therefore, it is quite conceivable that the cellular HSP protection system cannot compensate for the therapeutic NIPP effects at various physiological levels.

However, this review article also clearly shows that data on the interaction of NIPP and HSP systems are strictly limited. The available studies only provide a vague idea about the modulation of HSP after NIPP treatment. It must be considered as a limitation that exclusively jet and jet-like devices have been used for NIPP formation (Table 2). In biomedical studies, it has always been shown that NIPP properties and biomedical NIPP effects always depend on the technology being used for NIPP generation. In addition, HSP10, as a binding partner of HSP60, is involved in the folding and stabilization of client proteins, particularly those that are newly synthesized or damaged due to stress [147,148]. In addition to its chaperone activity, HSP10 has also been implicated in other cellular processes, including regulation of gene expression, cell growth, and apoptosis [149]. Aberrant expression or function of HSP10 has been associated with various diseases, including cancer, neurodegenerative disorders, and autoimmune diseases. However, there are no studies related to NIPP and HSP10 so far. Furthermore, the different biological model systems of the investigations hamper the comparison of the results. Here, systematic studies on the most important members of the HSP superfamily would be desirable in the future.

## Figures and Tables

**Figure 1 biomedicines-11-01471-f001:**
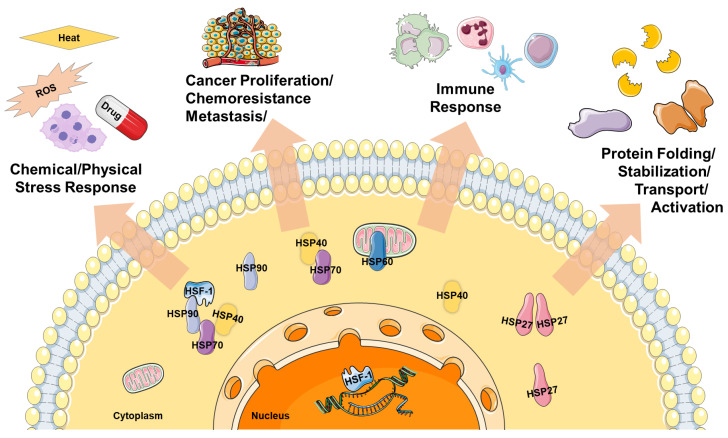
Heat shock proteins (HSP) are substantial cellular regulators of the eukaryotic stress response. HSP are named according to their molecular weights. The major members of the HSP superfamily are classified as HSP27, HSP40, HSP60, HSP70 and HSP90. Heat shock transcription factor 1 (HSF1) is considered the central factor of HSP expression. HSP mediate inducible protective mechanisms against chemical and physical noxae, including chemical and physical therapy approaches. HSP also exhibit cytoprotective and thus prooncogenic effects in tumor diseases and the interaction of cancer cells with the microenvironment and tumor-associated components of the immune system. At the molecular level, HSP control the correct folding of proteins, but also determine their stability and turnover, and thus the functionality of protein factors in cell physiology. For details, please refer to the text.

**Figure 2 biomedicines-11-01471-f002:**
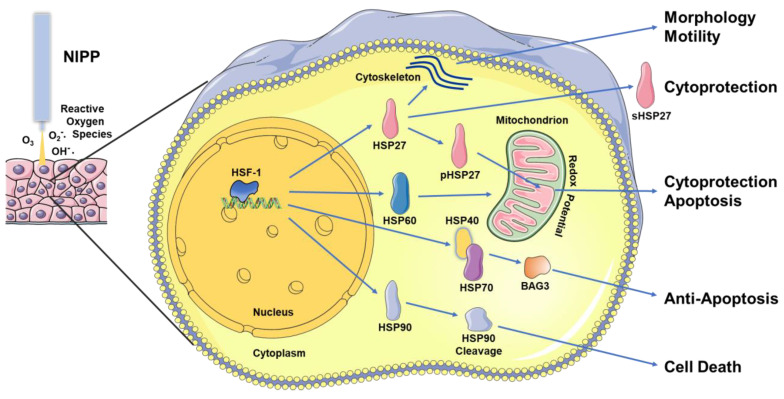
Non-invasive physical plasma (NIPP) treatment introduces reactive oxygen species (ROS) such as ozone (O_3_), superoxide anion (O_2_^−^•), and hydroxyl radical (OH^−^•) into the tissue. Within the cells, a stress-induced response of the heat shock protein (HSP) system occurs. Mediated by transcriptionally active heat shock factor-1 (HSF-1), HSP27 and HSP60 are induced, partially phosphorylated (pHSP27), and secreted (sHPS27), controlling cytoskeletal and mitochondrial functionality. Furthermore, large HSP70 and HSP90 and the co-chaperone HSP40 are activated. As part of the NIPP effect, HSP90 can also be cleaved and thus inactivated. Cell responses controlled by HSP include morphology, motility, cytoprotection and, depending on the redox state, apoptosis and cell death.

**Table 1 biomedicines-11-01471-t001:** Technical parameters, where published, of NIPP devices used in the studies to modulate HSP functionality after NIPP exposure.

Technology	Plasma Source	Voltage	Frequency	Airflow	Reference
Jet	Argon	2–6 kV	N/A	5 L/min	[59]
Jet	Helium	N/A	N/A	1/3/5 L/min	[60]
Jet	Argon + N_2_	18 kV	20 kHz	2 L/min	[61]
Jet	Argon	5 kV	10 kHz	3 L/min	[62]
Jet	Argon	N/A	N/A	N/A	[63]
DBD-based volume NIPP	Argon	7 kV	10 kHz	3 L/min	[64]
Jet	Argon	2–6 kV	1.1 MHz	3 L/min	[65]
Jet	N/A	N/A	N/A	N/A	[66]
Jet	Argon	N/A	1 MHz	4 L/min	[67]

**Table 2 biomedicines-11-01471-t002:** Overview of the non-invasive plasma physical (NIPP) devices used in research, the biological models used and which heat shock proteins (HSP) were characterized. Western blot (WB); polymerase chain reaction (PCR); immunofluorescence (IF); enzyme-linked immunosorbent assay (ELISA).

Technology	Model	HSP	Methodology	Reference
Jet	HaCat Cells	HSP27	WB	[59]
Jet	SaOS-2 Cells	HSP60	WB	[60]
Jet	U937 Cells	HSP40/70	PCR	[61]
Jet	Wistar Rat		IF	[62]
Jet	MDA-MB-s31	HSP90	WB	[63]
DBD-based volume CAP	Human Wound Skin		IF	[64]
Jet	THP-1 Cells	HSP27	WB	[65]
Jet	LNCaP/PC-3 Cells	HSP27	ELISA	[66]
Jet	OVCAR3 Cells	HSP27	ELISA	[67]

## Data Availability

Not applicable.

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
