# Peer review of "Impact of Non-Invasive Physical Plasma on Heat Shock Protein Functionality in Eukaryotic Cells"

_biomedicines, 2023, doi:10.3390/biomedicines11051471_

Round 1

Reviewer 1 Report

Comments:

Line 24 please correct the plasma definition. Various definitions can be found but basically a plasma is a gas with sufficient ionization level to display a collective response to EM fields. The presence radicals is peculiar of low temperature, chemically complex plasma, and I would skip EM radiation from the “mixture”.

Line 27-28, “the ionization process on avalanche charge multiplication by impact of electrons on atoms” or something similar, would read better.

Line 37 “trapped”: please consider a different formulation “stopped”, “arrested”, “cannot propagate freely due to…”

Line 130: typo “strongly”?

Section 7: conclusions. The authors states (correctly) at lines 304-305 that the biological response to NIPP is strongly dependent on the used technology. The term “jet “ is nonetheless misleading, referring just to the gasdynamic properties of the plasma source. It would be useful to include in table 1 and eventually in the discussion some information on the gas used and on the disharge excitation method – radiofrequecy, dielectric-barrier discharge, arc, corona (if used) . In fact, the plasma properties (and in partcular the ROS-RONS chemistry) are heavily dependent on these factors.

I Have no remarks on English quality

Reviewer 2 Report

The author made a systematic review and summary of Impact of Non-Invasive Physical Plasma on Heat Shock Protein Functionality in Eukaryotic Cells. The author discussed the relevant mechanism, but the author should further study the induction based on the characteristics of plasma.

The author can make targeted revisions for the following two issues:

1. Can the author add some key parameters dealing with cell plasma devices, such as device structure, voltage current, power, generated active particle density, etc.?

2. Since the author believes that the effect of plasma is limited, what is the threshold for triggering Heat Shock Protein Functionality in Eukaryotic Cell? How close is the plasma to triggering these thresholds?

There is no problem with the English expression of this paper.
